# Juvenile Osteochondral Lesions of the Talus: Current Concepts Review and an Update on the Literature

**DOI:** 10.3390/children10050884

**Published:** 2023-05-15

**Authors:** Albert T. Anastasio, Kian Bagheri, Emily M. Peairs, Caitlin Grant, Samuel B. Adams

**Affiliations:** 1Department of Orthopaedic Surgery, Duke University School of Medicine, Durham, NC 27710, USA; k_bagheri1204@email.campbell.edu (K.B.); emily.peairs@duke.edu (E.M.P.); caitlin.grant@duke.edu (C.G.); samuel.adams@duke.edu (S.B.A.); 2Campbell University School of Osteopathic, Lillington, NC 27546, USA

**Keywords:** osteochondral lesion of the talus, juvenile, microtrauma, ankle, osteoarthritis

## Abstract

Osteochondral lesions of the talus (OLTs) are lesions that occur before the physis closes and are frequently associated with acute ankle trauma. These lesions are often difficult to diagnose due to swelling and inflammation that are present after the initial injury. A growing body of literature has assessed the effects of OLTs in the adult population. However, the literature examining these lesions in the juvenile population is sparse. The purpose of this review is to provide a thorough understanding of OLTs, with a specific focus on the juvenile population. We evaluate the recent literature regarding the outcomes of various surgical treatment; modalities in the pediatric patient. While the outcomes after surgical treatment of pediatric OLTs are generally favorable, the paucity of investigation in this demographic is alarming. Further research is needed to better inform practitioners and families regarding these outcomes, as treatment plans are highly dependent on the individual patient in question.

## 1. Introduction

Osteochondral lesions of the talus (OLTs) are pathologic lesions of the talar cartilage and subchondral bone [1]. These lesions represent a challenge to orthopedic surgeons due to the avascular nature of the articular cartilage, which limits the capacity for self-repair and regeneration [2]. It has also been suggested that spontaneous healing in cases of OLTs is inhibited by microtraumas, which results in osteochondral injury and, in some cases, osteoarthritis (OA) over the long term [3]. 

Although OLTs often occur after ankle trauma, such as a sprain or fracture, Wang et al. estimated that up to 24% of patients cannot recall a specific cause of injury [4]. In many cases, the clinical presentation is asymptomatic. However, patients may experience mild, chronic pain accompanied by swelling, stiffness, or locking [3]. The pain is often described as being deeper compared to a traditional ligamentous injury of the ankle, with concomitant range-of-motion (ROM) abnormalities despite conservative treatment [4]. A classic scenario is a patient with trauma to the ankle that has persistent ankle pain after the resolution of the initial injury [5]. While ankle trauma is the most common cause of OLTs, other etiologies include cystic lesions [4], chronic overload due to malalignment [6], instability of the ankle joint [6], and even endocrine or metabolic causes [7]. It is generally accepted that the postoperative prognosis of OLTs is not affected by the etiology of the lesion; however, this area represents a focus of current research [7]. 

Management of OLTs involves both surgical and nonsurgical treatment options. Marrow stimulation has been long considered a first-line treatment, as it stimulates the subchondral plate to promote the growth of fibrocartilage [8]. However, this treatment is problematic in that the fibrocartilage formed from the procedure is biomechanically weaker than articular, hyaline cartilage [9,10]. Nevertheless, with recent advances in arthroscopic techniques and biological developments, the potential options for surgical management of OLTs are increasing [11]. 

While OLTs in the adult population have been well studied, little is known about the clinical efficacy of the different treatment options in the juvenile population [1]. Juvenile OLTs are lesions that occur before the physis closes. OLTs can be worrisome in the pediatric population due to the risk of progression to early OA [12]. Moreover, OLTs in patients who are closer to skeletal maturity can behave differently from those found in younger, less skeletally mature patients. Generally, chondral lesions in children are thought to resolve more readily than lesions found in adults [13]. However, newer research has demonstrated relatively poor healing rates in juvenile patients with OLTs, with complete radiographic healing noted in only 47% of lesions [14].

Given the paucity of high-quality evidence on OLTs in the juvenile population, the primary aim of this review is to give an overview of the pathogenesis of the disease, diagnostic workup, and treatment options, with a specific focus on the literature published from 2015 onward, with the goal of educating practitioners and improving patient outcomes for OLTs encountered in the juvenile population. 

## 2. Etiology and Epidemiology 

The current understanding of osteochondral lesions demonstrates varied etiology and is a point of continued investigation. When osteochondral lesions cause consequent articular cartilage abnormalities, they are termed osteochondritis dissecans (OCDs) [15]. OLTs are often characterized by osseous resorption, osseous collapse, and delamination of the cartilage [16]. It was previously thought that these lesions occur due to inflammation and subsequent necrosis, but histological studies of excised lesions have failed to demonstrate consistent osseous inflammation or any signs of widespread cellular necrosis [16,17]. Instead, findings of necrosis or inflammation in histological examination of OLTs are often dependent upon the state of attachment of the lesion and exact pathological mechanisms [16]. 

More recently, histological studies in animal models of OLTs in the knee demonstrated that poor blood supply and ischemia are implicated in lesion formation, especially in adolescence, as the subchondral vascular supply transitions from a perichondral source to a medullary cavity source [18,19,20,21]. Advanced imaging techniques with magnetic resonance imaging (MRI) have shown vascular regression in the epiphyseal cartilage during ossification, a finding which may portend increased susceptibility to OLTs in pediatric populations [22]. 

With regard to specific causes of OLTs, the most accepted and common etiology remains ankle trauma [6]. A recent systematic review of ankle fractures found an osteochondral lesion incidence of 45%, with nearly half of these lesions occurring in the talus [23]. The association of OLTs with trauma is mirrored in the juvenile population. In a case series of pediatric patients with OLTs, a history of preceding trauma was present in 79% of patients [24]. Additionally, competitive sports participation has been implicated in pediatric OLTs, with a reported 67.4% of these lesions occurring in patients participating in contact sports [25]. 

When examining the epidemiology of OLTs of the ankle in the pediatric population, studies have demonstrated a slight female predominance, with a female-to-male ratio of 1.6/1 [25]. This contrasts with the epidemiology of OLTs in the adult patient, with men being more commonly affected than women [26]. Moreover, juvenile OLTs appear to exhibit a far higher incidence in adolescent age groups as opposed to younger cohorts. Kessler et al. demonstrated a mean age of diagnosis for juvenile OLTs of 14 years, with no patients presenting under the age of 6 years and a 6.9 times greater risk in patients 12–19 years of age than those in the 6–11 year age group [25].

Regarding the specific location of cartilage lesions on the talus, the medial aspect of the talus is the most common location of OLTs post-ankle fracture, likely due to aberrant rotation and translation of the talus into the tibial plafond during a high-energy inversion injury [23]. These findings are mirrored in juvenile OLTs, with medial OLTs being more common than lateral. In adolescent OLTs, the medial aspect is the site of over 73% of lesions, with the lateral aspect representing upward of 22.4% of OLTs in this age cohort [24,25]. 

## 3. Diagnostics 

Osteochondral lesions of the ankle are often asymptomatic and may only be discovered incidentally on radiographs [18]. If the patient is symptomatic, symptoms typically include intermittent pain with weightbearing activities. A more severe lesion may lead to joint swelling, instability, locking, and intense pain, sometimes termed an “articular crisis” [27]. First-line diagnostics consists of a thorough history and physical, with a focus on previous ankle injuries, localized tenderness, ROM, and ligamentous laxity [18]. 

After a history and physical, first-line imaging for ankle OLTs is a radiographic examination. Radiographs in the anterior–posterior (AP) direction or with the ankle in 15° of internal rotation (mortise view) are particularly useful. On the AP radiograph, it is sometimes possible to see a “subchondral halo”, while the mortise view may be useful for evaluating the superior and lateral corners of the talus [18,24,28]. Advanced imaging options can yield greater detail about the location and severity of the lesion, as well as evaluate the underlying cartilage. Computed tomography (CT) provides more precise information about the lesion location compared to radiographs, but MRI is preferred for assessing the underlying integrity of the cartilage [29]. CT arthrography can also be helpful in assessing both the lesion size and cartilage quality but involves the use of both ionizing radiation and intra-articular injections in the adolescent population. These risk exposures must be carefully considered along with any potential diagnostic benefit of CT arthrography [29].

## 4. Treatment

Like most orthopedic pathologies, treatment for juvenile OLTs is guided by both patient factors and the severity of the disease. For OLTs that are not detached or free-floating, conservative treatment of leg immobilization and protective weightbearing is recommended [30,31,32]. Younger patients such as those under the age of 10 are more likely to have more open physes. These patients have greater healing potential and may benefit more from conservative treatment than adults and older adolescent patients who are closer to skeletal maturity [24,33,34]. As such, in most cases of juvenile OLTs, conservative treatment is often first-line. A typical conservative treatment protocol involves immobilization and no weightbearing, with or without nonsteroidal anti-inflammatory drugs (NSAIDs) for roughly 6 weeks, followed by progressive weightbearing and physical therapy, which often includes soft-tissue massage, joint mobilization, and exercises to improve flexibility, strength, and balance [35].

Unfortunately, it is estimated that more than one-third of patients with OLTs will fail conservative treatment [35]. For patients who fail conservative treatment or have detached and free-floating lesions, surgical intervention is indicated. In most cases, several surgical treatment options are viable; thus, the treatment choice is dependent on the size and type of defect, in addition to patient preference. 

Orthopedic surgeons are taught several classification schemas for OLTs, which consider the varied presentation of the condition and help guide treatment. The Berndt and Harty radiographic classification is the most frequently used and separates OLTs into four stages [26,36]. In Stage 1, radiographic findings show evidence of a small area of subchondral compression. In Stage 2, there is partial fragment detachment. In Stage 3, there is complete fragment detachment but no displacement. In Stage 4, the fragment is completely displaced. Lesions that are Stage 1 or 2 and small Stage 3 lesions are all generally treated conservatively at first; large Stage 3 and any Stage 4 lesions are considered operative candidates [35]. 

Surgical options include debridement of the necrotic subchondral bone, removal or internal fixation of the fragment, and bone marrow stimulation, as mentioned above. Tissue transplantation techniques such as osteochondral allograft (e.g., particulated juvenile allograft cartilage (DeNovo Graft, Zimmer Biomet; Warsaw, Indiana), Figure 1, Figure 2 and Figure 3) or autograft and autologous chondrocyte implantation are also options for restoring the articular surface and preventing further degenerative change and progression to OA. 

## 5. Literature Review

Given that greater than half of the boys and one-quarter of girls in the 8–16 year age range engage in some form of competitive, organized sport during the school year, it follows that children commonly expose the ankle joint to repetitive microtraumas, leading to a relatively high incidence of OLTs, particularly in adolescent cohorts [37]. A vast body of literature has been published on the management of OLTs in the adult population. To date, however, very few studies have focused specifically on the management of juvenile OLTs. Here, we discuss the recent literature regarding OLTs in the juvenile population. We conducted a careful and comprehensive literature review utilizing the following databases: PubMed, Embase, and Scopus. Manuscripts regarding OLTs published since 2015 in juvenile patients were extracted. We excluded items that included only an abstract and no full manuscript. Manuscripts written in languages other than English were also excluded. Our aim is to provide an update on the current state of OLT management in pediatric populations and give providers a sense of the outcomes they can expect with various treatment modalities. Results of the recent studies on juvenile OLT are summarized in Table 1.

A retrospective case series by Ikuta et al. assessed the clinical outcomes of retrograde drilling (RD) for OLT in juvenile patients [38]. This study included eight juvenile patients (five boys and three girls) with a mean follow-up of 2 years. The authors utilized the American Orthopedic Foot and Ankle Society (AOFAS) score and ankle activity score to evaluate the clinical outcomes. Arthroscopic findings were also graded according to the International Cartilage Research Society (ICRS) classification system. The AOFAS score significantly improved from 69.3 (59.6–78.9) to 97.1 (93.3–100.9) postoperatively (*p* = 0.012). The ankle activity score also improved from 2.0 (1.6–2.5) to 6.6 (5.5–7.8) postoperatively (*p* = 0.011). Each participant returned to athletic activity at their pre-symptomatic level of performance 6 months after surgery. OLTs of ICRS grade 0 and 1 were identified arthroscopically in three and five patients, respectively. The authors concluded that RD is a potential option for treating juvenile patients with OLTs refractory to nonoperative treatment at short-term follow-up. Further work reporting long-term outcomes of RD in juvenile patients at larger sample sizes would greatly enhance our understanding of the durability of this procedure, particularly regarding whether this treatment modality can decrease the rate of subsequent posttraumatic arthritis. 

A case series by Carlson et al. evaluated the functional and radiographic outcomes for children and adolescents undergoing arthroscopic treatment of symptomatic OLTs with marrow-stimulating techniques [15]. These authors described their algorithm for surgical decision making during arthroscopic evaluation of juvenile OLTs. Once the lesion is identified, the size and stability of the defect are assessed through direct visualization and utilization of the arthroscopic probe. In the setting of a stable lesion (defined by these authors as a lesion where cartilage is determined to be intact after probing), transtalar drilling is initiated under fluoroscopic guidance. The cartilage overlying the lesion is left intact. In the setting of a loose or unstable lesion (a lesion that can easily be dislodged during probing or where the lesion hinges about a tenuous attachment site), lesion excision is carried out. Subsequently, transtalar drilling is again performed, followed by the addition of microfracture in the lesion bed to generate bleeding bone for deposition of fibrocartilage. 

Outcomes regarding this described procedure for juvenile OLTs are generally favorable. The minimum follow-up time for this study was 2 years. The study group consisted of 22 patients (11 male and 11 female) with a mean age of 14.4 years (8–18 years) and a mean follow-up of 8.3 years (2–27 years). Outcome measures for this study included the Foot Function Index (FFI), AOFAS, Tegner Activity Scale (TAS), Short-Form 36 (SF-36), and visual analog scale (VAS). At the end of this study, 20 of 22 patients were satisfied with their results. The overall mean AOFAS score at follow-up was 86.6. The mean postoperative FFI score overall was 38.7, while the mean SF-36 physical component score was 50.7. The mean TAS score changed from 7.2 preoperatively to 6.0 postoperatively. The overall mean follow-up score for the visual analog scale for pain was 2.2 on a 10-point scale. 

These authors also reported lesion filling in outcomes utilizing the postoperative MRI magnetic resonance observation of cartilage repair tissue (MOCART) score. The mean MOCART score was 48.0 in this series. Complete filling of the cartilage occurred in 27% of cases while complete graft integration occurred, and an intact repair surface was noted in 22% of the cases. The authors concluded that the arthroscopic treatment of symptomatic OLTs in adolescent patients demonstrated excellent overall outcomes, as evidenced by high clinical satisfaction rates and improvements in functional outcomes. The addition of transtalar drilling to microfracture and lesion excision in the setting of unstable OLTs may further enhance healing. 

Other studies have proposed cartilage regenerative techniques for the treatment of juvenile OLTs. A retrospective case series by Körner et al. assessed patient-reported outcome (PRO) measures after combined matrix-associated autologous chondrocyte implantation (MACI) and autologous bone grafting in high-stage OLTs in adolescents [39]. A total of 12 adolescent patients (13 ankles) were included in this study, four male and nine female cases, with a mean age at the time of surgery of 17.7 ± 2.1 years. The median follow-up for this study was 80 months (range 22–107 months). The authors analyzed clinical efficacy with the Foot and Ankle Outcome Score (FAOS) and Foot and Ankle Ability Measure (FAAM). The mean overall FAOS score was 78 ± 13. The results for the subscales of FAAM were as follows: activities of daily living, 81 ± 20, function/activities of daily living, 84 ± 13, sports, 65 ± 29; function/sports, 73 ± 27. On the basis of these results, the study concluded that PRO measures following MACI and autologous bone grafting for high-stage OLTs in adolescents are mixed. Further work will be required before MACI and other cartilage regenerative techniques can be recommended for the treatment of juvenile OLTs. However, this avenue may represent an area of significant potential, as MACI has been shown to result in chondrocyte differentiation into articular hyaline cartilage, with similar aggrecan, type II collagen, and S-100 expression [46]. Future work should aim to assess whether type II hyaline collagen deposition after MACI in lieu of the fibrocartilage which is deposited after microfracture [47] leads to reduced onset of arthritis at long-term follow-up. 

Another study by the same lead author (Körner et al.) analyzed the reoperation rate after surgical treatment of OLTs in children and adolescents [40]. This study included 27 consecutive patients with a solitary OLT (10 male and 17 female) with a mean age of 16.9 ± 2.2 years. These patients received primary operative treatment via arthroscopy and bone marrow stimulation (BMS) (*n =* 8), arthroscopy and retrograde drilling (*n =* 8), autologous chondrocyte implantation and autologous bone grafting (n = 9), arthroscopy, BMS, and retrograde drilling (*n =* 1), and flake fixation (*n =* 1). Of the 27 patients, seven required reoperation (reoperation rate of 25.9%) after a median interval of 31 months. Patients with reoperation had significantly lower ICRS classification stages compared to patients without reoperation. While further work will be required to comment on the durability of the heterogenous set of procedures described in this series, providers should be aware of a relatively high reoperation rate in patients undergoing surgical treatment for juvenile OLTs and should counsel patients appropriately. 

The recent literature points toward generally favorable outcomes for the treatment of OLTs in juvenile populations, albeit with relatively high reoperation rates. However, a dearth of literature regarding long-term outcomes of juvenile OLTs clouds the current understanding of the efficacy of these treatments in preventing perhaps the most important sequela of OLTs: progression to OA. Furthermore, the quality of evidence for most of the studies in the current literature is lacking. Many of the studies were retrospective cohort studies with a small number of patients. Thus, we must stress the importance of future, high-quality research with larger patient populations to more adequately understand this critical adolescent pathology. 

Further work is also required to guide the creation of specific treatment algorithms for these lesions. For example, age and degree of skeletal maturity greatly impact the prognosis for OLT. Should surgeons be more aggressive in the treatment of OLTs in patients who are nearing skeletal maturity? While this logic is theoretically sound, we currently lack the long-term data to definitively conclude which lesions will benefit from surgical treatment in the prevention of posttraumatic arthritis or talar collapse. Thus, further efforts geared specifically toward our understanding of juvenile OLT will greatly enhance our ability to provide evidence-based care for children with OLT. In particular, we call for a focus on prognostic indicators of lesion progression and the creation of specific treatment protocols tailored to the unique situations in which OLT can present in children.

## 6. Conclusions

In summary, juvenile OLTs (lesions that occur before the physis closes) are relatively common in the adolescent population, particularly athletes. In contrast to the adult population, less is known about the optimal treatment modality and outcomes for OLTs in children. The literature published since 2015 on OLTs in juvenile patients has demonstrated a favoring of more aggressive treatment for OLT in these cohorts, with generally favorable results in patients treated with surgery. However, given the ongoing paucity of the literature specifically evaluating the unique behavior of juvenile OLTs and the proposed therapeutic options, we urge additional high-quality studies that involve larger patient populations to provide prognostic details and guidance for the conservative or surgical management of juvenile OLT. 

## Figures and Tables

**Figure 1 children-10-00884-f001:**
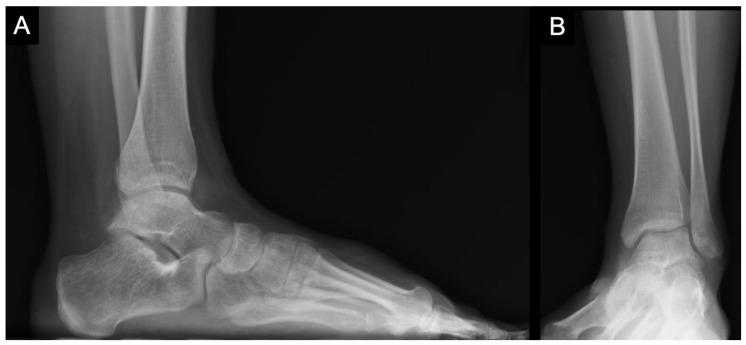
(**A**) Preoperative radiographic imaging of a 14-year-old female (with distal tibial physis nearing complete closure) with evidence of OLT at the medial talar dome. (**A**) Lateral radiograph of the left foot demonstrates a lucent lesion with minimal sclerosis, most consistent with OLT. (**B**) Mortise view of the left ankle reveals the subtle OLT at the medial talar dome.

**Figure 2 children-10-00884-f002:**
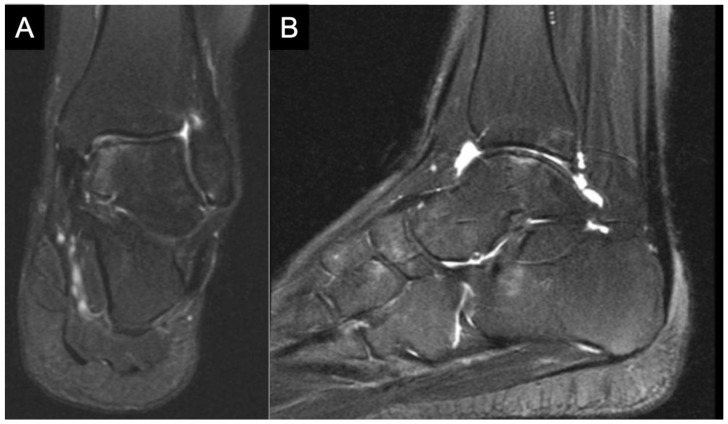
(**A**) Preoperative MRI in a 14-year-old female reveals further characteristics of the OLT. (**A**) Coronal MRI view reveals subchondral bone marrow edema. (**B**) Sagittal MRI view further defines the OLT, with associated bone marrow edema and joint effusion. After obtaining the MRI, given ongoing symptomology in the setting of OLT with associated lateral ligamentous instability, the decision was made to proceed with ankle arthroscopy, application of particulated juvenile allograft cartilage implant (DeNovo Graft, Zimmer Biomet; Warsaw, Indiana), and lateral ligamentous stabilization through a modified Bostrom procedure.

**Figure 3 children-10-00884-f003:**
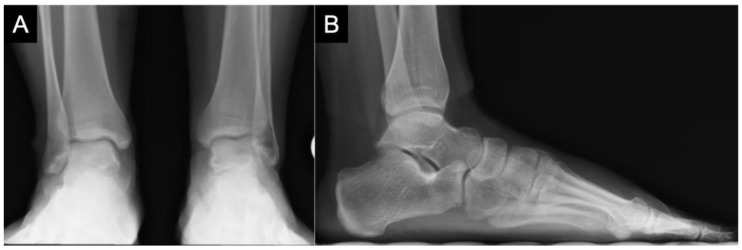
(**A**) Postoperative radiographic imaging of an 14-year-old female (**A**) Anteroposterior imaging reveals ongoing remodeling of OLT in the left medial talar dome. (**B**) Lateral radiograph reveals subtle improvement in subchondral cystic change indicative of remodeling.

**Table 1 children-10-00884-t001:** A summary of recent studies published on various treatment modalities for juvenile OLT.

Study	Design	Number of Participants	Male-to-Female Ratio	Mean Age (Years)	Mean Follow-Up (Years)	Intervention	Outcome	Description of Results
Ikuta et al., 2020 [38] *	Retrospective case series	8	5:3	14.9	2	RD	Benefit	The mean total ROM was 65.6° preoperatively and 67.5° postoperatively (*p* = 0.55). The AOFAS score improved from 69.3 to 97.1 postoperatively (*p* = 0.012). The ankle activity score improved from 2.0 to 6.6 postoperatively (*p* = 0.011).
Carlson et al., 2020 [15]	Retrospective case series	22	11:11	14.4	8.3	Arthroscopy with marrow stimulation	Benefit	Of 22 patients, 20 were satisfied with their results and would recommend it to others. The overall mean follow-up score for the VAS for pain was 2.2. The overall mean AOFAS score at follow-up was 86.6.
Körner et al., 2021 [39]	Retrospective case series	12	4:9	17.7	6	MACI with autologous bone grafting	Heterogeneous results	The mean overall FAOS score was 78 ± 13. The FAOS subscale scores were as follows: *symptoms*, 70 ± 14; *pain*, 83 ± 10; *activities of daily living*, 89 ± 12; *sports/recreational activities*, 66 ± 26; *quality of life*, 51 ± 17.
Körner et al., 2021 [40]	Retrospective case series	27	10:17	16.9	3.5	Arthroscopy + BMS, *n* = 8; arthroscopy + RD, *n* = 8; ACI/autologous bone grafting, *n* = 9; arthroscopy + BMS + RD, *n* = 1; flake fixation, *n* = 1	Heterogenous results	The reoperation rate was 25.9% (7 of 27 patients). Patients with reoperation had significantly lower ICRS classification stages compared to patients without reoperation.
Heyse et al., 2015 * [41]	Retrospective comparative study	67 patients (77 lesions)	27:40	11.4	5.8	Conservative treatment	Inferior	Every patient but one received conservative treatment initially. Overall, 61% of lesions failed conservative treatment. Increased age and grade 3 lesions at diagnosis were predictive for failure of conservative treatment. Higher-grade lesions were generally predictive of inferior outcomes.
Jurina et al., 2018 * [42]	Case series	13	7:6	15	5.6	Arthroscopic microfracture	Benefit	According to Berndt and Harty outcomes, good clinical results were reported in 10 (76.9%) patients, and fair clinical results were reported in 3 (23.1%) patients. There was a statistically significant improvement in the postoperative AOFAS score compared to the pre-operative AOFAS score, with a mean increase of 35 points.
Pagliazzi et al., 2018 * [43]	Retrospective review	7	1:6	12.8	4.0	Arthroscopic BMAC	Benefit	Six lesions were Stage 3 according to the Berndt and Harty classification and 1 lesion was Stage 4. There was a statistically significant postoperative improvement in AOFAS score compared to preoperative, with a mean increase of 36.9 points. VAS scores significantly improved from 6.3 preoperatively to 0.4 at final follow-up.
Masquijo et al., 2016 * [44]	Retrospective chart review	6	5:1	13	3.1	Arthroscopic RD	Heterogenous results	All patients were asymptomatic at final follow-up; however, only 3 out of 6 of them had a complete radiographic healing at last follow-up. The average AOFAS score significantly improved from 69 points preoperatively to 98 points postoperatively. VAS scores significantly improved from 6.2 preoperatively to 0.3 postoperatively.
Minokawa et al., 2020 * [45]	Retrospective case series	6 patients (8 ankles)	4:2	11.1	1.9	RD	Benefit	The mean JSSF scale in all ankles improved significantly from 79.4 points preoperatively to 98.4 points at final follow-up. Final follow-up CT findings showed that 4 ankles demonstrated good healing, 3 were fair, and 1 was poor.
Dahmen et al., 2022 [1]	Systematic review	381 lesions (20 studies)	213:168	13	4.9	Conservative treatment, *n* = 8 studies; BMS, *n* = 8 studies; RD, *n* = 6 studies; fixation, n = 4 studies	Heterogenous results	The mean MINORS score of the included studies was 7.6. The pooled success rate was 44% in the conservative group, 77% in the BMS group, 95% in the RD group, 79% in the fixation group, and 67% in the autograft group.

Table legend. RD, retrograde drilling; ROM, range-of-motion; AOFAS, American Orthopedic Foot and Ankle Society; VAS, visual analog scale; MACI, matrix-associated autologous chondrocyte implantation; FAOS, Foot and Ankle Outcome Score; BMS, bone marrow stimulation; ACI, autologous chondrocyte implantation; ICRS, International Cartilage Repair Society; BMAC, bone marrow aspirate concentrate; JSSF, Japanese Society for Surgery of the Foot; CT, computed tomography; MINORS, Methodological Index for Nonrandomized Studies. * These studies were all included in the systematic review by Dahmen et al. (2022).

## Data Availability

No new data were created or analyzed in this study. Data sharing is not applicable to this article.

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
