# Peer review of "Juvenile Osteochondral Lesions of the Talus: Current Concepts Review and an Update on the Literature"

_children, 2023, doi:10.3390/children10050884_

Round 1

Reviewer 1 Report

I'd like to thank the authors for this paper.

In general, this paper is well-written and covers most aspects of the OCL topic.

Though, in my opinion, this paper adds little to the general knowledge. There are several very similar papers that have been published, and most have also been cited by this one. The main importance of this paper is to show how little is known, and the low quality of evidence, specifically in pediatric patients. This fact by itself may be important enough to warrant publication, both to help practitioners discuss treatment options with patients as well as encourage further research in the field. 

I have a few remarks that I believe may improve the manuscript:

1. There is a long discussion that OCLs are primarily caused by trauma. Some statistics are given to support this. Though, no discussion is present about the other causes - what are other causes? Is there a difference in treatment/prognosis in these cases? Is / how the OCL is different in pediatric vs adolescent vs adult patients - all is mentioned very shortly, please elaborate.

2. A discussion of the quality of evidence is warranted. Most of the research papers mentioned are retrospective cohorts with a small number of patients. This fact should be stressed and discussed to encourage further higher-quality research into this field.

3. The paragraph about the typical locations of the OCL (lines 89-93) is not very clear. Please rewrite and clarify.
4. Please elaborate on the conservative treatment, an example of conservative protocol would be useful. This would be the first-line treatment for many of the patients and it is possibly one of the most practical things this article can provide to clinicians.

5. In references, all journal article titles end with "of the article", please remove.  

Reviewer 2 Report

Dear Author,

Thank you for the opportunity to review this article.

The introduction is too scarce. Is there any classification of OCD that you may use when talking about the treatment?

You could elaborate more on the treatment.

A Prisma flow diagram should be useful when constructing a literature review. What are the criteria for choosing those 5 studies? Does the systematic review of 20 studies (Dahmen et el.) already include the previous 4 studies in its paper? You need to clarify your decision to choose only these studies as you already talk about another manuscript with a lot more sources than yours.

The review should have a clear conclusion, if any. Please rewrite it, because it looks rather like a Discussion.

Minor editing of English language required

Round 2

Reviewer 1 Report

Thank you for addressing the concerns raised in the previous version.

I'm happy with the current version and have no further suggestions.

Reviewer 2 Report

Thank you for including our comments and adding value to the article.